# Brief Communication: AI-driven rapid landslides mapping following the 2024 Hualien City Earthquake in Taiwan

Lorenzo Nava[1, 2, 3 *], Alessandro Novellino[4, *], Chengyong Fang[5], Kushanav Bhuyan[5,3],
Kathryn Leeming[4], Itahisa Gonzalez Alvarez[4], Claire Dashwood[4], Sophie Doward[4], Rahul Chahel[4],
Emma McAllister[4], Sansar Raj Meena[3], and Filippo Catani[3]

[1]Department of Earth Sciences, University of Cambridge, Cambridge, UK
[2]Department of Geography, University of Cambridge, Cambridge, UK
[3]Machine Intelligence and Slope Stability Laboratory, Department of Geosciences, University of Padova, 35129 Padua, Italy
[4]British Geological Survey, Nicker Hill, Keyworth, NG12 5GG, United Kingdom
[5]State Key Laboratory of Geohazard Prevention and Geoenvironment Protection, Chengdu University of Technology, 610059, Chengdu, China
[*]These authors contributed equally to this work.
**Correspondence:** Lorenzo Nava (ln413@cam.ac.uk)

**Abstract.**

On April $2^{nd}$, 2024, a Mw 7.4 earthquake struck Taiwan's eastern coast, triggering numerous landslides and severely impacting infrastructure. To create a preliminary inventory of the earthquake-induced landslides in Eastern Taiwan ( 3,300 km$^2$) we deployed automated landslide detection methods by combining Earth Observation (EO) data with Artificial Intelligence models. The models identified 7,090 landslide events covering >75 km$^2$, in $\approx$ 3 hours after the acquisition of the EO imagery. This research showcase AI's role for rapid landslide detection for disaster response. The generated landslide inventory can also be used to improve the understanding of earthquake-landslide interactions to improve seismic hazard mitigation.

## 1 Introduction

Taiwan is prone to high landslide hazards due to frequent rainfall and earthquake events (Hung, 2000; Chuang et al., 2021; Shou and Chen, 2021). A significant portion of Taiwan's population and its infrastructure are vulnerable to these hazards (Lee and Fei, 2015). On 2nd of April 2024, Taiwan was hit by a Mw 7.4 earthquake (United States Geological Survey - USGS, 2024). The shaking resulted in a large number of landslides along transport routes with >1,100 people injured (https://disasterphilanthropy. org/disasters/2024-taiwan-earthquake/). A complete and up-to-date landslide inventory is important as a support during the emergency response (Amatya et al., 2023) and also for a better understanding of the spatio-temporal relationships between landslide occurrence and driving factors (Lombardo et al., 2020). Such information can redefine triggering thresholds for landslide early warnings and hazard zoning for land use planning.

Over the last decades, spaceborne Earth Observation (EO) has become a predominant source for mapping landslides, which are particularly useful to first responders (Amatya et al., 2023; Novellino et al., 2024). Mapping landslides using Earth Observation (EO) data has become crucial for providing vital situational awareness to first responders during large-scale landslide

events. Recently, there have been significant advances in AI-based automated landslide detection and mapping (Novellino et al., 2024). These approaches include utilizing crowdsourced data (Catani, 2021) and Unmanned Aerial Vehicles (UAVs) (Dai et al., 2023), as well as analyzing LIDAR (Fang et al., 2022) and satellite optical imagery (Amatya et al., 2021; Bhuyan et al., 2023), and SAR (Nava et al., 2022).

Additionally, there is a growing trend toward training deep learning (DL) models capable of providing reliable predictions in new areas for rapid assessment of widespread multiple landslide events (MLEs). We find studies focusing on a single data source, such as Copernicus Sentinel-2 (Prakash et al., 2021) and PlanetScope (Meena et al., 2023), while others investigate the integration of multisource data (Fang et al., 2024; Xu et al., 2024) to enhance accuracy and improve transferability.

Despite this large amount of research, there remains a scarcity of real-world applications leveraging AI techniques in new, unseen large landslide events. Currently, to our best knowledge, Amatya et al. (2023) stand out as one of the few studies where automatic landslide mapping methods were applied as part of disaster response activities following the 2021 earthquake in Haiti. However, as areas and methods change, more investigation of such applications as well as AI-based methods must be undertaken to speed up the trust and understanding of how such automated systems can efficiently improve disaster response.

In this Brief Communication, we test in practice state-of-the-art AI techniques on different EO satellite data for the automatic detection and mapping of landslides associated with the event. We further provide suggestions about how these tools can support future rapid landslide mapping efforts following major disasters worldwide. Lastly, we provide the preliminary co-seismic landslide inventory for updating landslide hazard models and supporting resilience to future events.

## 2   Hualien City earthquake and study area

On the 2$^{nd}$ of April 2024 (23:58 UTC), a Mw 7.4 earthquake struck the eastern coast of Taiwan (USGS, 2024). The event was located at a depth of 40km with an epicentre near the town of Hualien (Figure 1)

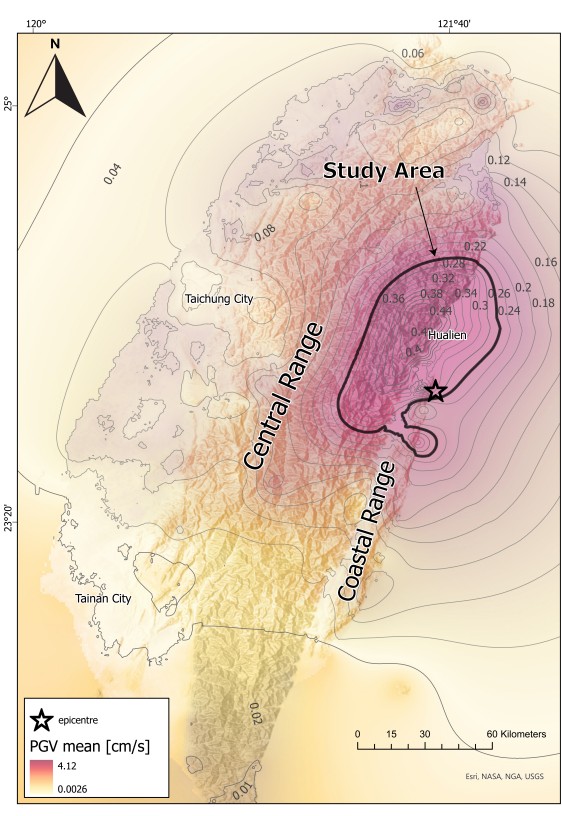

**Figure 1.** Peak Ground Velocity (PGV) values, Peak Ground Acceleration (PGA) contours and epicentre for the Hualien City earthquake (from USGS, 2024). The 0.2%g is in black bold and represents the area of study of this work. Sources: Esri, DeLorme, HERE, TomTom, Intermap, increment P Corp., GEBCO, USGS, FAO, NPS, NRCAN, GeoBase, IGN, Kadaster NL, Ordnance Survey, Esri Japan, METI, Esri China (Hong Kong), swisstopo, MapmyIndia, and the GIS User Community.

as a result of a reverse NE-SW fault near the boundary between the Eurasian and Philippine Sea plates. The main earthquake was followed by a Mw 6.5 aftershock 13 minutes later. Eastern Taiwan is not only tectonically active but is also relentlessly

battered by hurricanes, making this location particularly prone to the rapid erosion of the mountain chains built by tectonics. Following information about the earthquake epicentre and effect (PGA) and reports on landslides from social media through the Global Landslide Detector (Pennington et al., 2022), we defined a 3,300 km$^2$ area of interest (AoI) for mapping landslides centred around the town of Hualien (> 0.2% PGA). The extent of the AoI is a trade-off between the extent of the shaking and the availability of cloud-free images in the aftermath of the event.

## 3  Automated Landslide Detection and Mapping

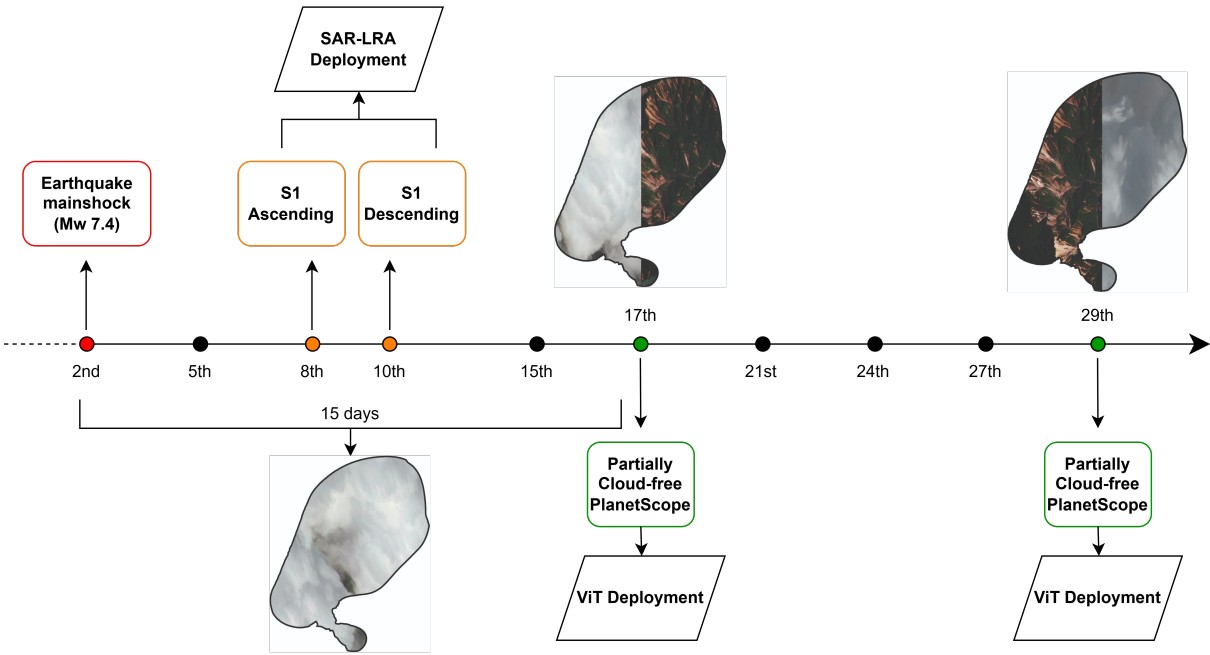

**Figure 2.** Timeline of satellite image acquisitions and models deployment in April 2024.

The landslide maps have been generated using the Synthetic Aperture Radar (SAR) Landslide Rapid Assessment (SAR-LRA) tool based on Convolutional Neural Networks (Nava et al., 2024) and a Vision Transformer (ViT) model (Tang et al., 2022; Fang et al., 2024).

The SAR-LRA tool was trained and validated on 11 MLEs globally distributed and uses pre- and post-event SAR imagery
in a change-detection-like approach to identify surface changes due to co-seismic slope failures. No transfer learning or fine-tuning was necessary; the model was directly deployed in the area. The tool is freely available at https://doi.org/10.5281/zenodo.14898556. SAR-LRA was applied over five Sentinel-1 acquisitions at 10m resolution. This included one acquisition on April 8, 2024, for the ascending geometry (over two different tracks), five SAR acquisitions within 60 days preceding the event,

and one acquisition on April 10, 2024, for the descending geometry. SAR data enabled landslide detection even under cloudy conditions, which prevented the use of optical Sentinel-2 data for several weeks post-earthquake (see Figure 2). SAR-LRA led us to identify preliminary hotspots of landslide-related surface changes.

The ViT model was pre-trained and validated on a multi-source landslide segmentation dataset (Fang et al., 2024), the Globally Distributed Coseismic Landslide Dataset (GDCLD). The GDCLD dataset integrates multi-source remote sensing imagery, including PlanetScope, Gaofen-6, Map World, and UAV data, covering landslides triggered by nine MLEs across diverse geological and geomorphological settings worldwide. Since AI models map spectral reflectance, their performance is influenced by the contrast between landslide-affected areas and their surroundings. Given that most landslides in GDCLD occur in densely vegetated areas, similar to Hualien, we expect the model to generalize well in this context. The GDCLD is available at https://doi.org/10.5281/zenodo.11369484 (Fang et al., 2024). We fine-tune the model (Bhuyan et al., 2023) on 814 landslides manually mapped within the Taiwan study area affected by the 2024 earthquake. These landslides were mapped across the affected area rather than all of Taiwan, and no specific landslide features were pre-selected. However, we included some negative samples (e.g., riverbeds and bare land) to improve model generalization (the subset is available in Supplementary Materials). Satellite images from the Google Earth Pro archive have been used for the pre-event stage, whose collection includes data from CNES and Airbus acquired up to September 2023. For the post-event stage, ViT has been applied to 33 composited PlanetScope images at 3 m spatial resolution acquired on the 17th and 29th of April, 2024.

## 4 Results and Discussion

We retrieved a total of 7090 co-seismic landslides along with 2,617 pre-seismic ones. SAR-LRA outputs 262 SAR-LRA bounding boxes: 63 in the ascending geometry and 199 in the descending geometry (Figure 3a). The co-seismic landslides encompass new failures, reactivations and/or remobilizations of existing landslides (Figures 3b-c). Most co-seismic slope failures occurred on slopes between 30 and 50 degrees on the SE slopes (Figure 3d). The total co-seismic landslide area resulting from the earthquake equals 75.3 km$^2$ with an individual polygon minimum size set to 250 m$^2$, due to the resolution of Planet images, up to a maximum of 2.9 km$^2$ (Figure 3e).

SAR-LRA yielded results in $\approx$ 20 minutes, while ViT analysis, including both pre- and post-processing tasks, took about 2 hours. This allowed us to produce co-seismic inventories within hours of satellite image acquisition. The SAR-LRA tool was fundamental in initially identifying landslide locations, as cloud cover was persistent for $\approx$ 15 days after the event.

Reflecting on our methodology, our initial concerns regarding the suitability of SAR imagery for Taiwan's steep slopes were alleviated by its successful validation in cloud-free areas. The initial skepticism likely stemmed from the visual characteristics of SAR data, which makes it difficult for the human eye to confirm the presence of the landslide predicted by the AI model. As complete cloud coverage over an entire region is rare, the SAR-based predictions could be partially validated using the landslides visible on optical. This step can increase the trustworthiness of our rapid assessment models. Regarding the optical-based predictions, after model fine-tuning, the results were generally reliable, with few false positives in flat areas that were easily masked out manually. The advantage of this approach is that we get the exact extent of the landslides. However, since

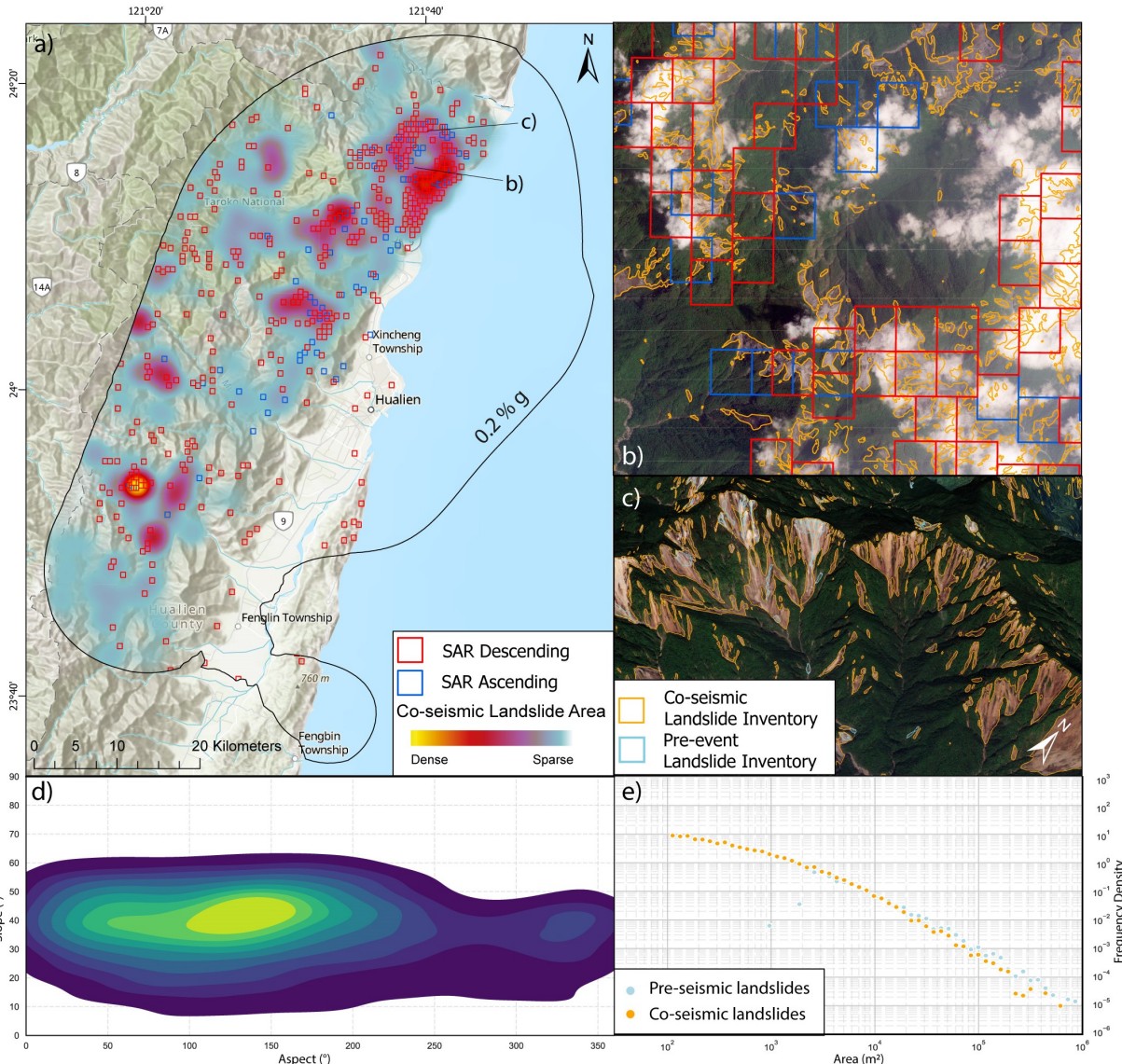

**Figure 3.** Overview of the landslide inventory (a). A zoom of the co-seismic landslides mapped with squares of SAR-LRA and the polygons of ViT (b-c). Density plot of slope vs aspect for the co-seismic landslides (d). Frequency area distribution of pre- and co-seismic landslides (e). Sources: Esri, DeLorme, HERE, TomTom, Intermap, increment P Corp., GEBCO, USGS, FAO, NPS, NRCAN, GeoBase, IGN, Kadaster NL, Ordnance Survey, Esri Japan, METI, Esri China (Hong Kong), swisstopo, MapmyIndia, and the GIS User Community. Map data ©2024 Google.

our approach relied solely on post-event imagery, we had to deploy the model also on pre-event imagery and subtract the two

inventories to identify the co-seismic landslides. Reflecting on this, approaches that integrate change-detection mechanisms within a single model are preferable and advocated.

Validating AI-based landslide detection during an emergency is challenging due to the lack of an immediate ground-truth inventory for comparison. To validate our inventory, we conducted a visual inspection of pre- and post-event PlanetScope imagery, which allowed us to confirm that detected landslides corresponded to actual surface changes. This process also helped us correct minor errors, particularly where the AI model slightly overestimated landslide extents or merged nearby landslides. We also analyzed the Frequency-Area Distribution (FAD) exponent of our co-seismic inventory and compare it with those from

other earthquake-triggered landslide inventories. Landslide size distributions typically follow a power-law relationship, with exponents $\approx$ 2-3 for seismic events. Our AI-derived exponent (2.0) aligns well with values reported for previous earthquakes triggered MLEs, including Gorkha 2015 (2.15, Roback et al., 2018), Papua New Guinea 2018 (2.04, Tanyas et al., 2022), and Wenchuan 2008 (2.13, Fan et al., 2018). This consistency suggests that our AI-mapped inventory captures a realistic landslide size distribution.

Overall, when performing automated landslide mapping in new events, we need to maximize the chances our AI-model will predict landslides accurately. To do so, transfer learning and/or fine-tuning a generalized model within the affected area is a well-established approach that significantly improves AI model performance in new regions (Bhuyan et al., 2023). This allows us to assume that the model will perform reliably despite the absence of immediate field validation. Additionally, checking FAD exponents serves as a further control to ensure that anomalous detections are minimized. Lastly, while AI-based predictions

provide a rapid mapping solution, a semi-automated approach remains preferable. Double-checking AI results with manual verification using pre- and post-event imagery will continue to be necessary to refine outputs and improve accuracy.

    Since available, we compared our AI-based inventory with the one published by Chen et al. (2025). Their inventory identified 1,243 landslides, whereas our has $\approx$ 7,000. While there is overlap between many polygons in the two inventories, our approach mapped many more landslides. Chen et al. noted that cloud cover and resolution limitations likely led to an underestimation of

125 smaller landslides. Additionally, the FAD rollover point (computed as the most frequent landslide size) is significantly lower in the AI-based inventory ($\approx$ 342.5 m$^2$ vs. $\approx$ 2,345 m$^2$ in the manual inventory), confirming that AI effectively detects smaller landslides. However, this also introduces well-known artifacts, such as amalgamation (merging of adjacent landslides) and fragmentation (splitting of single landslides), as observed in previous studies (Bhuyan et al., 2023).

## 5   Conclusions

Following the Hualien City earthquake event, we semi-automatically map $\approx$ 7,090 co-seismic landslides from satellite imagery at different resolutions and different data modalities using AI-based approaches. While there is a wealth of literature on the use of AI for landslide detection, there are few documented cases of its application for rapid mapping in the aftermath of major disasters. This research makes two primary contributions. First, we demonstrate and evaluate the application of AI for rapid landslide assessment in disaster response. Specifically, we highlight how the SAR-based automated approach (SAR-LRA

Tool) played a crucial role in accurately identifying landslide locations despite persistent cloud coverage. In contrast, optical

data, while offering higher precision, became available only after significant delays. Second, we provide an open-source inventory that delivers essential information for situational awareness, aids emergency responders during disaster aftermath, and facilitates the updating of landslide hazard models, thereby enhancing resilience to future events. Overall, given the demonstrated effectiveness of these approaches and tools, we are confident that they can be successfully deployed in future large-scale earthquake-triggered landslide events, provided that manual quality checks are implemented. Integrating SAR and Optical AI approaches will further improve the reliability and performance of rapid assessment models, especially in challenging weather conditions. These advancements will provide disaster responders with valuable information in future MLEs.

*Code and data availability.* The generated inventory and the subset used to fine-tune the ViT is freely available on Zenodo at the link: https://zenodo.org/records/11519683. The code and weights of SAR-LRA tool is available at https://github.com/lorenzonava96/SAR-and-DL-for-Landslide-Rapid-Assessment/tree/main. The Globally Distributed Coseismic Landslide Dataset (GDCLD) is available at https://doi.org/10.5281/zenodo.11369484. Planet imagery can be found at https://www.planet.com/. Sentinel-1 imagery can be found in the Copernicus Data Space Ecosystem at https://dataspace.copernicus.eu/.

*Author contributions.* LN and AN: conceptualization, data curation, analysis, visualization, and writing– original draft. CF and KB: data curation, analysis, writing– original draft, writing– review, and editing. KL, IGA, CD, SD, RC, EM, SRM: writing– review, and editing. FC: writing– review, editing, and funding acquisition.

*Competing interests.* At least one of the (co-)authors is a member of the editorial board of Natural Hazards and Earth System Sciences.

*Disclaimer.* (will be included in the published version of the article)

*Acknowledgements.* The authors would like to thank E. Hussain from the British Geological Survey who provided comments to improve the quality of this work. We also thank the anonymous reviewers for their valuable suggestions and insightful feedback, which significantly improved this manuscript. The research was supported by the ALMEO project, part of the BGS 'Geoscience to tackle Global Environmental Challenges' grant (NE/X006255/1) and by the "The Geosciences for Sustainable Development" project (Budget Ministero dell'Università e della Ricerca–Dipartimenti di Eccellenza 2023–2027 (C93C23002690001).

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
