# Peer review of "Brief Communication: AI-driven rapid landslides mapping following the 2024 Hualien City Earthquake in Taiwan"

_Natural Hazards and Earth System Sciences, 2024_

## Author Response (AR1)

*The authors sincerely thank the Editor and the anonymous reviewers for their valuable suggestions and constructive feedback. We greatly appreciate the time and effort dedicated to improving this manuscript.*

Reviewer #1

RC: *The authors presented a brief communication about the sinergic use of AI remote sensing tools, based on radar and optical imagery, and their implementation as a potential rapid mapping tool to be impleme for emergency response. The paper is very interesting and promising in its goal, as well as well presented. I have some minor comments which I hope will help its comprehension:*

AR: Thank you for your feedback and suggestions. We answer to your comments point by point as follows:

RC: *Would you mind to improve the overall quality and readability of Figure 1? the study area boundaries and its topography are barely visible as it is. Figure 2, it would be more effective to use the sum of the days after the triggering of the events in the timeline, instead of the dates.*

AR: We agree and we will improve the figure according to your suggestions. Specifically, in Figure 1 we improved the transparency and thickness of the study area boundary, and added "Study Area" in the image. In Figure 2 we added the sum of days without optical images, along with the dates to give a clearer overall picture.

RC: *line 25: what LMEs stand for?*

AR: We amended and added the extended version: Multiple landslide events (MLEs)

RC: *you used the GDCLD database to train the model over Taiwan case of study. It would be more insightful to add some description of such database. Are these landslides comparable, in terms of geological geomorphological and land use setting, geometric and topographic features, to those triggered in Taiwan? Please, specify it.*

AR: Thank you for your comment. We agree that providing more details on the GDCLD database would improve clarity. The GDCLD integrates multi-source remote sensing imagery, including data from PlanetScope, Gaofen-6, Map World, and uncrewed aerial vehicles (UAVs). It covers landslides triggered by nine multiple landslide events (MLEs) across different geological and geomorphological settings worldwide, specifically in Jiuzhaigou, Hokkaido, Mainling, Palu, Mesetas, Nippes, Sumatra, Lushan, and Luding. These sites encompass diverse land use patterns, geometric characteristics, and topographic conditions. Regarding the comparability of these landslides to those in Taiwan, we acknowledge that the geological information in the GDCLD is limited. However, in terms of geomorphology, both the GDCLD dataset and the Taiwan case study primarily focus on mountainous terrains, which share similarities in their topographic features. Moreover, since our AI model primarily maps spatial features based on spectral reflectance, its performance is influenced by the contrast between landslide-affected areas and their surroundings. In densely vegetated regions, landslides typically exhibit distinct spectral signatures due to vegetation removal. Given that the majority of landslides in the GDCLD dataset occur in densely vegetated areas, as is the case in the Hualien MLE, we can expect the model to generalize well in this context. We thank you again and will clarify these points in the manuscript to enhance transparency regarding the dataset and its applicability to the Taiwan case study. Added in lines 75 to 82.

RC: *You mentioned that 814 landslides were manually mapped. Are these landslides mapped in a specific area of Taiwan or allover the country? Did you focus on specific landslides or features?*

AR: Thank you for your question. We manually annotated 816 landslides using PlanetScope imagery following the Hualien earthquake. These landslides were mapped across the affected area rather than all of Taiwan, and no specific landslide features were pre-selected. However, we included some negative samples (e.g., riverbeds and bare land) to improve model generalization. We will release this portion of the data as supplementary materia and we will add this information to the manuscript. Added in lines 84 to 86.

RC: *I would like to see some lines about how to validate these landslides. Validation is completely missing here. Please, consider to add some remarks on how to validate landslides mapped by automatic tools.*

AR: We thank you and agree with this comment. This is also aligned to a suggestion of reviewer 2.

We applied the following approaches to evaluate our inventory:

i) We conducted a qualitative validation by visually inspecting pre- and post-event PlanetScope imagery. This allowed us to confirm that detected landslides corresponded to actual surface changes and correct minor errors, particularly where the AI model slightly overestimated or merged nearby landslides.

Following your comments, we have now added the following validation steps:

ii) we computed and compared FAD exponents with those from similar earthquake-triggered landslide inventories. Landslide size distributions typically follow a power-law relationship, with exponents around ∼2 for similar seismic events. Our AI-derived exponent (∼2.0) aligns well with previously observed values, including: 2015 Gorkha, Nepal: 2.15 (Roback et al., 2018); 2018 Papua New Guinea: 2.04 (Tanyas et al., 2022); 2016 Kaikoura, New Zealand: 2.45 (Tanyas et al., 2022b); 2008 Wenchuan, China: 2.13 (Fan et al., 2018). This consistency suggests that our model captures a realistic distribution of landslide sizes.

iii) We compared our AI-based inventory with the Chen et al. (2025) inventory and found a significant difference in mapped landslide numbers (Chen: 1,243; AI-based: ∼7,000). While there is overlap in certain mapped landslides, Chen et al. noted that cloud cover and resolution constraints likely led to an underestimation of smaller landslides. Our AI model's pre-event prediction on Google imagery identified 2,611 landslides, while the co-seismic AI prediction on PlanetScope imagery (after subtracting pre-event landslides) mapped 7,090 landslides larger than 250 m². To further investigate these differences, we analyzed the power-law exponents (α) and rollover points of the inventories. While our AI-derived exponent remains close to 2.0, Chen's manual dataset exhibits a steeper exponent (2.43), confirming the underrepresentation of smaller landslides suggested by the authors. The rollover point, computed as the most frequent landslide area, is also much lower in the AI-based inventory (∼342.52 m² vs. ∼2,345 m² in the manual inventory), reinforcing that the AI model detects smaller events more effectively. However, this also introduces well-known challenges, such as amalgamation (where multiple adjacent landslides are merged) and fragmentation (where a single landslide is split into multiple patches). These artifacts are typical of pixel-based AI models, as observed in previous studies (Bhuyan et al., 2023). Please see the figures at the end of the comment for additional visual insights.

However, validating AI-based landslide detection during an emergency is inherently difficult due to the lack of an immediate ground-truth inventory for comparison. In such cases, alternative validation strategies must be considered. Transfer learning and fine-tuning a generalized model within the affected area is a well-established approach that significantly improves AI model performance in new regions (Bhuyan et al., 2023). This allows us to reasonably assume that the model will perform reliably despite the absence of immediate field validation. Additionally, checking FAD exponents serves as a further control to ensure that anomalous detections are minimized. Lastly, while AI-based predictions provide a rapid mapping solution, a semi-automated approach remains preferable. Double-checking AI results with manual verification using pre- and post-event imagery will continue to be necessary to refine outputs and improve accuracy.

We have added this validation steps and considerations at lines 120 to 143.

**Reviewer #2**

*RC: The rapid extraction of landslides after the earthquake is an important task of post-earthquake relief and disaster assessment, and this manuscript carries out the post-earthquake landslide extraction of 2024 Hualien City Earthquake in Taiwan based on AI technology, which is an interesting work.*

AR: Thank you for your feedback and suggestions. We answer to your comments point by point as follows:

*RC: The author said in the manuscript that no landslide inventory 15 for the 2024 Hualien City earthquake has been released, in fact, there are a number of articles on the landslide after the 2024 Hualien City Earthquake in Taiwan has been published.*

AR: Thank you for your comment. You are correct that landslide inventories for the 2024 Hualien earthquake do exist, and we have identified two relevant studies: Chang et al. (2024) and Chen et al. (2025). However, only the inventory from Chen et al. is openly available. Please find a comparison between our AI-based and Chen et al.'s inventories in the answer to your 3rd comment, and in the manuscript at lines 137 to 143.

RC: *page2: what DL and LMEs stand for?*

AR: We amended and added the extended version: Multiple landslide events (MLEs).

RC: *How to verify the accuracy of landslide extraction after the 2024 Hualien City Earthquake in Taiwan based on AI technology?*

AR: We thank you for your comment. This is also aligned to a comment of reviewer 1.

We applied the following approaches to evaluate our inventory:

i) We conducted a qualitative validation by visually inspecting pre- and post-event PlanetScope imagery. This allowed us to confirm that detected landslides corresponded to actual surface changes and correct minor errors, particularly where the AI model slightly overestimated or merged nearby landslides.

Following your comments, we have now added the following validation steps:

ii) we computed and compared FAD exponents with those from similar earthquake-triggered landslide inventories. Landslide size distributions typically follow a power-law relationship, with exponents around ~2 for similar seismic events. Our AI-derived exponent (~2.0) aligns well with previously observed values, including: 2015 Gorkha, Nepal: 2.15 (Roback et al., 2018); 2018 Papua New Guinea: 2.04 (Tanyas et al., 2022); 2016 Kaikoura, New Zealand: 2.45 (Tanyas et al., 2022b); 2008 Wenchuan, China: 2.13 (Fan et al., 2018). This consistency suggests that our model captures a realistic distribution of landslide sizes.

iii) We compared our AI-based inventory with the Chen et al. (2025) inventory and found a significant difference in mapped landslide numbers (Chen: 1,243; AI-based: ~7,000). While there is overlap in certain mapped landslides, Chen et al. noted that cloud cover and resolution constraints likely led to an underestimation of smaller landslides. Our AI model's pre-event prediction on Google imagery identified 2,611 landslides, while the co-seismic AI prediction on PlanetScope imagery (after subtracting pre-event landslides) mapped 7,090 landslides larger than 250 m². To further investigate these differences, we analyzed the power-law exponents ($\alpha$) and rollover points of the inventories. While our AI-derived exponent remains close to 2.0, Chen's manual dataset exhibits a steeper exponent (2.43), confirming the underrepresentation of smaller landslides suggested by the authors. The rollover point, computed as the most frequent landslide area, is also much lower in the AI-based inventory (~342.52 m² vs. ~2,345 m² in the manual inventory), reinforcing that the AI model detects smaller events more effectively. However, this also introduces well-known challenges, such as amalgamation (where multiple adjacent landslides are merged) and fragmentation (where a single landslide is split into multiple patches). These artifacts are typical of pixel-based AI models, as observed in previous studies (Bhuyan et al., 2023). Please see the figures at the end of the comment for additional visual insights.

However, validating AI-based landslide detection during an emergency is inherently difficult due to the lack of an immediate ground-truth inventory for comparison. In such cases, alternative validation strategies must be considered. Transfer learning and fine-tuning a generalized model within the affected area is a well-established approach that significantly improves AI model performance in new regions (Bhuyan et al., 2023). This allows us to reasonably assume that the model will perform reliably despite the absence of immediate field validation. Additionally, checking FAD exponents serves as a further control to ensure that anomalous detections are minimized. Lastly, while AI-based predictions provide a rapid mapping solution, a semi-automated approach remains preferable. Double-checking AI results with manual verification using pre- and post-event imagery will continue to be necessary to refine outputs and improve accuracy.

We have added this validation steps and considerations at lines 120 to 143.

REFERENCES

Tanyaş, H., Görüm, T., Fadel, I. et al. An open dataset for landslides triggered by the 2016 Mw 7.8 Kaikōura earthquake, New Zealand. Landslides 19, 1405–1420 (2022). https://doi.org/10.1007/s10346-022-01869-9

Chang, JM., Chao, WA., Yang, CM. et al. Coseismic and subsequent landslides of the 2024 Hualien earthquake (M7.2) on April 3 in Taiwan. Landslides 21, 2591–2595 (2024). https://doi.org/10.1007/s10346-024-02312-x

Chen, Y., Song, C., Li, Z. et al. Preliminary analysis of landslides induced by the 3 April 2024 Mw 7.4 Hualien, Taiwan earthquake. Landslides (2025). https://doi.org/10.1007/s10346-025-02465-3

Bhuyan, K., Tanyaş, H., Nava, L. et al. Generating multi-temporal landslide inventories through a general deep transfer learning strategy using HR EO data. Sci Rep 13, 162 (2023). https://doi.org/10.1038/s41598-022-27352-y

Fan, X., Juang, C. H., Wasowski, J., Huang, R., Xu, Q., Scaringi, G., ... & Havenith, H. B. What we have learned from the 2008 Wenchuan Earthquake and its aftermath: A decade of research and challenges. Engineering Geology, 241, 25-32 (2018). https://doi.org/10.1016/j.enggeo.2018.05.004